# Effects of Magnesium Oxide and Magnesium Hydroxide Microparticle Foliar Treatment on Tomato PR Gene Expression and Leaf Microbiome

**DOI:** 10.3390/microorganisms9061217

**Published:** 2021-06-04

**Authors:** Aggeliki Andreadelli, Spyros Petrakis, Antiopi Tsoureki, George Tsiolas, Sofia Michailidou, Penelope Baltzopoulou, Robert van Merkestein, Philip Hodgson, Mark Sceats, George Karagiannakis, Antonios M. Makris

**Affiliations:** 1Institute of Applied Biosciences, Centre for Research & Technology, Hellas (CERTH), 570 01 Thessaloniki, Greece; a.andreadelli@certh.gr (A.A.); spetrak@certh.gr (S.P.); adatsoureki@certh.gr (A.T.); george.tsiolas@certh.gr (G.T.); sofia_micha28@certh.gr (S.M.); 2Chemical Process & Energy Resources Institute, Centre for Research & Technology, Hellas (CERTH), 570 01 Thessaloniki, Greece; pbaltzop@cperi.certh.gr (P.B.); gkarag@certh.gr (G.K.); 3Calix Limited, Pymble, NSW 2073, Australia; rvmerkestein@calix.global (R.v.M.); phodgson@calix.global (P.H.); mgsceats@calix.global (M.S.)

**Keywords:** MgO nanoparticles, Mg(OH)_2_ nanoparticles, porous micron particles (PMP), Mg(OH)_2_ adhesion, plant signalling, leaf microbiome, antibacterial, crop protection

## Abstract

Recently, metal oxides and magnesium hydroxide nanoparticles (NPs) with high surface-to-volume ratios were shown to possess antibacterial properties with applications in biomedicine and agriculture. To assess recent observations from field trials on tomatoes showing resistance to pathogen attacks, porous micron-scale particles composed of nano-grains of MgO were hydrated and sprayed on the leaves of healthy tomato (*Solanum lycopersicum*) plants in a 20-day program. The results showed that the spray induced (a) a modest and selective stress gene response that was consistent with the absence of phytotoxicity and the production of salicylic acid as a signalling response to pathogens; (b) a shift of the phylloplane microbiota from near 100% dominance by Gram (−) bacteria, leaving extremophiles and cyanobacteria to cover the void; and (c) a response of the fungal leaf phylloplane that showed that the leaf epiphytome was unchanged but the fungal load was reduced by about 70%. The direct microbiome changes together with the low level priming of the plant’s immune system may explain the previously observed resistance to pathogen assaults in field tomato plants sprayed with the same hydrated porous micron-scale particles.

## 1. Introduction

The efficient use of fertilizers, pesticides, and fossil fuels is necessary to reduce the carbon footprint of production and damage to the environment [1]. A recent study [2] estimated crop yield loss by pests and pathogens to range between 17 and 23% for four crops (wheat, maize, potato, and soybean), while the estimate was 30% for rice. Specifically, for tomato, where crop losses are due to more than 200 pests and diseases, a recent review provided data for the most common diseases with 10–80% yield reductions [3]. No major changes in crop health occurred between previous studies in 2001–2003 and 2017, when the latest global survey on crop losses was conducted. Bacteria, fungi, and viruses are major pathogen contributors. Recently, metal oxides (TiO_2_, ZnO, CuO, MgO, etc.) and magnesium hydroxide (MgOH)_2_ nanoparticles (NPs) with high surface-to-volume ratios were shown to possess antibacterial properties, and these materials are finding applications in biomedicine and agriculture [4,5,6,7,8,9]. In the case of (Cu)-based antimicrobials, which are currently widely used by the agricultural sector for disease control, the trend is to minimize their usage due to potential toxicity and reduced effectiveness in several cases (EPA-HQOPP-2010-0212).

Thus, there is a pressing need to explore and develop effective, non-persistent and non-bioaccumulative alternatives, and NPs are a class of such materials that are being explored [6]. NPs are organic, inorganic, or hybrid particles with at least one of their dimensions ranging from 1 to 100 nm (at the nanoscale). Nanoparticles offer the advantage of the effective delivery of agrochemicals due to their large surface area, easy attachment, and fast mass transfer [10]. Metal oxide nanomaterials, such as CuO, ZnO, MgO, and aluminium oxide have been shown in laboratory tests to control many foliar and soilborne plant diseases caused by *Botrytis cinerea, Alternaria alternate, Monilinia fructicola, Colletotrichum gloeosporioides, Fusarium solani, Fusarium oxysporum* f.sp. *radicis-lycopersici**, Verticillium dahliae, Phytophthora infestans*, and *Ralstonia solanacearum* in many plant species [11,12,13,14]. There are differing models of the mode of action of NPs, and these are not fully resolved. The increased stability of agrochemicals, antifungal, antibacterial properties (possibly through the disruption of membrane stabilization and reactive oxygen species (ROS)), pesticidal properties, and positive contributions to stress tolerance are some of the NP modes of action discussed [14].

Magnesium is an essential nutrient for living organisms. In plants, most Mg^2+^ is associated with proteins, with cytosolic Mg^2+^ present at approximate concentrations of 0.4 mM. An additional 15–20% is associated with chlorophyll pigments, acting mostly as a cofactor of enzymes involved in photosynthetic carbon fixation and metabolism [5,15]. Magnesium deficiency can result in shorter and smaller roots, necrotic spots on leaves due to impaired carbon metabolism, and photosynthesis [16]. Compounding this, the demands of agricultural production to cope with the increasing human population have resulted in decreased concentrations of magnesium in wheat seeds. Among inorganic metal oxides, magnesium oxide nanoparticles (MgO NPs) have been shown to be antibacterial agents with the advantages of being non-toxic, non-persistent, and non-bioaccumulative. While the United States Food and Drug Administration classifies MgO and Mg(OH)_2_ as safe [17] and the European Chemical Agency has concluded that Mg(OH)_2_ is not persistent, bioaccumulative, or toxic, the NP form of these materials has not been explicitly mentioned.

As inorganic ionic antimicrobial agents, MgO NPs have been found to exhibit a direct and wide spectrum of toxicity against several types of pathogens, such as Gram-negative (*E. coli*) and Gram-positive (*S. aureus*) bacteria [5], as well as the fungi *Aspergillus niger* and *Penicillium oxalicum* [18]. For example, commercially available MgO NPs were tested by Cai et al. against the Gram-negative, soil-borne *Ralstonia solanacearum*, a pathogen which infects numerous plant species [5]. These nanoparticles were analysed by SEM and found to have a surface area consistent with a spherical particles with particle diameters in the range of 50–100 nm. In liquid suspensions, the MgO NPs could completely kill bacteria at concentrations of 250 μg/mL. To assess the effects of MgO NPs on the morphology of *R. solanacearum* cells, they examined the microbial cells by SEM. Incubated pathogen cells exhibited deep craters on their surface, and many of them had burst. Interestingly, even control standard bulk MgO had some effects on bacteria, creating small holes on their surface. These visual changes corresponded with increases in efflux of pathogen DNA and proteins into the aqueous solution, and a loss of viability was confirmed by flow cytometry. Staining with DCF, an ROS-sensitive dye, indicated that MgO NPs contributed to ROS production. The NP suspension in water is expected to hydrate to a degree, as well as produce ROS that are able to directly penetrate cells and cause death [5]. Previous investigations have indicated that MgO NPs induce systemic resistance within tomato plants by activating salicylic acid (SA), jasmonic acid (JA), and ethylene signalling pathways [19]. Magnesium hydroxide (Mg(OH)_2_) NPs also appear to possess antimicrobial properties [20,21]. *E. coli* and *Pseudomonas syringae* were especially susceptible to elimination by (Mg(OH)_2_) NPs. When tested by spraying on plants, no phytotoxicity was evident by visual observation, in contrast to CuSO_4_ application [6].

A challenge to the widespread use of such NPs is the complexity and cost of the synthesis of NPs at the commercial scale. Recently, the use of porous, micron-scale particles, herein called PMPs, was developed [22] as a means of delivering the same bioactivity as NPs of the same chemical composition. The availability of commercial quantities of material have allowed for field trials over many years, and the results of field trials of flash-hydrated MgO-PMPs were recently published [23]. In that work, hectare-scale field trials using flash-hydrated MgO PMPs were carried out at commercial tomato farms and were managed according to integrated pest management (IPM) protocols. The foliar application of PMP suspensions resulted in about 20 μg/cm^2^ leaf coverage to elicit either a reduction in damaged fruit, associated with reduced insect pest pressure or a reduction in the frequency at which pest pressure reached economically damaging levels. Such a PMP treatment required substantially less insecticide and fungicide compared to control plots, which were farmed using conventional crop protection actives alone [23].

In the current work, we assessed the impact of spraying aqueous suspensions of variably hydrated MgO PMPs onto healthy tomato plants. Plant leaf samples were analysed to determine whether any changes in plant stress response genes and the profile of the leaf microbiome coincided with the treatments. The work focused on changes to the healthy plant and its phyllosphere from spraying, prior to pathogen attack, as the first step in understanding the mode of action of MgO PMP-based materials on pathogens and pests. The proposed link between the bioactivity of PMPs in field trials and laboratory experiments on equivalent NPs is that the grains of PMPs are on the nanoscale. NPs have a tendency to agglomerate into weakly bound clusters that may emulate PMPs, thus supporting the hypothesis that the bioactivity of both NPs and PMPs is linked to their respective characteristic nanostructures.

## 2. Materials and Methods

### 2.1. Material

The MgO PMP was prepared from the ground mineral magnesite, MgCO_3_, with a particle size distribution in the range of 0.3–90 μm in a 25-m-long drop reactor with a wall temperature of about 1050 °C to induce the calcination of the MgCO_3_ at a temperature not exceeding 900 °C and to form MgO PMP through the loss of CO_2_ to create lattice vacancies in a short time so as to minimize sintering. There was very little decrepitation of the MgCO_3_ particles during this process, with the particle size distribution of the product being similar to that of the ground MgCO_3_ raw material.

The flash-calcination process allowed for the rapid removal of CO_2_ from the carbonate structure, leading to highly porous micron-scale particles with high surface area and reactive properties similar to those of nanoparticles (NPs) but without the safety concerns and high production cost typically associated with nano-materials [22,24].

### 2.2. Physico-Chemical Characterization Methods

For MgO PMP characterization, the following techniques were employed: small angle X-ray scattering (Australian Synchrotron, Clayton, VIC, Australia); X-ray diffraction—XRD (Siemens D500/501 diffractometer using Cu Kα radiation, λ = 1.5406 Å); scanning electron microscopy coupled with energy dispersive X-ray spectroscopy—SEM/EDS (JSM-6300 JEOL); transmission electron microscopy (Phillips CM2000) in which a dilute suspension of the MgO powder samples in absolute ethanol were dispersed and a drop was pipetted onto a copper grid before imaging, according to manufacturer’s instructions in several replicate samples; and thermo-gravimetric analysis—TGA (SETARAM SETSYS Evolution 18 Analyzer with Al_2_O_3_ crucibles) in which the measurement was performed under air-flow (100 mL/min) with a heating rate of 5 °C/min to achieve well-controlled oxidation conditions, according to authors’ measurement protocol. Moreover, the particle size distribution of the sample was determined by laser diffraction in 10 replicate samples (Malvern Mastersizer 3000, Spectris, Surrey, England, UK) in pure ethanol, while the specific surface area and pore volume measurements were carried out by liquid nitrogen adsorption/desorption isotherms (Quantachrome^®^ ASiQwin™-Version 2.02, Boynton Beach, FL, USA) using BET and BJH analysis, respectively.

X-ray diffraction traces were obtained from powdered samples with a Panalytical XPert Powder Diffractometer (Malvern Panalytical, Spectris, Surrey, England, UK) using CoK_α_ radiation and graphite monochromator. Operating conditions were 40 kV/30 mA, a step scan of 0.02θ/2θ at 1°/2θ/min, 1° of divergence and receiving slits, and a 0.15° scatter slit. The scan range was 5–76° 2θ. Phases were identified by a computer search/match of the ICDD PDF4 2019 Database. Quantitative results were determined using HighScore Plus software and the most suitable structures; HighScore is a whole pattern Rietveld technique.

### 2.3. Suspension Preparation and Analysis

For the field trials previously reported in [23], the MgO PMP was flash-hydrated to give near 100% hydrated Mg(OH)_2_ particles in a stable 55–60 wt% solid suspension. The temperature during hydration rose to about 90 °C from the heat of hydration. The cooled suspension was transported to the farm and diluted with water in an approximately 1:20 mass ratio and sprayed to give the leaf coverage in excess of the coverage used in the field trials, estimated to be at 2.2 μg/cm^2^ [20]. For the laboratory work reported herein, the process of the hydration of the MgO PMP was varied to allow for research into the role of the hydration process in bioactivity, with a particular emphasis on determining the effect, if any, of the temperature rise in the flash-hydration process.

The suspensions used in the current work for spraying were prepared by mixing deionized water with the aforementioned MgO PMP powder. The amount of MgO used in all cases was 2.3 wt%. It is noted that for MgO in water at such concentrations, the equilibrium favours the near complete conversion into Mg(OH)_2_. However, in practice, the low solubility of both MgO and Mg(OH)_2_ particles in water allows for the formation of materials with a mix of MgO and Mg(OH)_2_. If hydration was complete, this would correspond to approximately 2.9 wt% Mg(OH)_2_ in the final suspension. Three different suspensions were prepared for application on the tomato plants. PMP-1 was prepared a few minutes (<3 min) before its application to the tomato plants as described below, PMP-2 was prepared at ambient temperature with continuous stirring for approximately 20 h, and PMP-3 was prepared at 90 °C with continuous stirring for approximately 20 h. PMP-3 was formally equivalent to the diluted flash-hydrated suspension sprayed in past field trials [23].

For a quantitative estimation of the rate and the degree of MgO hydration, additional samples identical to the abovementioned suspensions were prepared and the hydration was measured as described in Section 3.2. For each of these samples, the solids were filtered and dried at 110 °C, and the degree of hydration was subsequently estimated by the weight loss recorded by TGA in the temperature range from ~200 to 450 °C where the dehydration of Mg(OH)_2_ typically occurs. The formation of Mg(OH)_2_ was also confirmed via the XRD analysis of the dried powders.

### 2.4. Plant Spraying and Sample Collection

In order to evaluate the effect of the application of active materials onto plant leaves, tomato plants (*Solanum lycopersicum* L.) of the Optima variety were chosen to be tested (Spring). Four groups of tomato plants with three biological replicates were used for spraying: one for each of the three active materials and one for the untreated plants used as control. The plants were sprayed with the already described suspensions of PMP-1, PMP-2, and PMP-3, as well as with deionized water for the control plants, until they were fully wetted. The experiments were repeated in the fall period with another commercial variety, Enza Zaden Elpida F1, that was suitable for the season. The plants were sprayed with PMP-2 or with sterile deionized water. Leaf samples prior to spraying were also collected as controls.

The plants were 40 cm tall, left outdoors for 3 days prior to treatment, and remained outdoors for the whole 20-day duration of the experiments. On the fifth day after spraying, researchers wearing gloves and using forceps collected tissue samples for gene expression analysis, as well as for metagenomics analysis. More precisely for the gene expression analysis, a single entire leaf from each of the three plants of each group was collected and placed in a 1.5 mL tube, flash-frozen in liquid nitrogen, and stored in −80 °C until RNA extraction. Additionally, for the metagenomics analysis, an entire single leaf of each tomato plant was selected from the plants and placed into a 50 mL sterile falcon tube with 40 mL of isotonic phosphate-buffered saline (PBS). After incubation for 1 h at room temperature in an orbital shaker, plant tissue was removed and the solution was centrifuged for 10 min in an IEC FL40 Centrifuge (ThermoFisher Scientific, Waltham, MA, USA) equipped with an M4 High Throughput Swing-Out Rotor and adapters for 50 mL conical tubes. For the collection of the microbial cells, the maximum speed of 4000 rpm (3970× *g*) was used. The supernatant was discarded, and the remaining material was stored at −20 °C until DNA extraction. Using this procedure, based on the sample collection notes of ZymoBIOMICS DNA Miniprep Kit, interference from mitochondrial and chloroplastic DNA within the plant genome on the determination of microorganisms is avoided. In the replicate experiment, for the “unsprayed (1)” and “dH_2_O (1)” samples, an SL1 Plus Thermo equipped with a TX-400 swing bucket rotor was used (4696× *g*).

### 2.5. DNA Extraction, Library Construction and Sequencing

Microbial DNA was extracted using the ZymoBIOMICS DNA Miniprep Kit (ZYMO RESEARCH; Irvine, CA, USA) according to the manufacturer’s instructions. DNA concentration was measured on a Qubit 4.0 Fluorometer using the Qubit^®^ dsDNA HS assay kit (Invitrogen, Carlsbad, CA, USA). Bacterial diversity was assessed by sequencing the V3–V4 region of the 16S rRNA gene, whereas fungal diversity was assessed by amplifying the V7–V8 region of the 18S rRNA gene. Libraries were constructed using Illumina’s 16S Metagenomic Sequencing Library Preparation (15044223 B) protocol. For the amplification of the V3–V4 region of the 16S rRNA gene, gene-specific primers were selected based on work of Klindworth et al., whereas for the amplification of the V7–V8 region of the 18S rRNA gene, the universal primers FR1 and FF390 were selected from the work of Chemidlin et al. [25]. All primers were modified by adding an Illumina (Illumina Inc., San Diego, CA, USA) overhang adapter nucleotide sequence at the 5’ end. The sequences of the primers used are shown in Appendix A. PCR products and libraries were purified to remove unincorporated primers and primer-dimer species using a NucleoMag^®^ NGS Bead Suspension (Macherey-Nagel, Düren, Germany). All libraries were quantified with fluorometric quantification using the Qubit^®^ dsDNA BR assay kit, and their size was evaluated on a Fragment Analyzer system (Agilent Technologies Inc., Santa Clara, CA, USA) using the DNF-477-0500 kit. The molarity of libraries was assessed by a qPCR conducted on a Rotor-Gene Q thermocycler (Qiagen, Hilden, Germany) with the KAPA Library Quantification kit for Illumina sequencing platforms (KAPA BIOSYSTEMS, Woburn, MA, USA). Libraries were sequenced on a MiSeq platform using the MiSeq^®^ reagent kit v3 (2 × 300 cycles) (Illumina, San Diego, CA, USA). Sequencing data were deposited in NCBI at BioProject ID PRJNA638533.

### 2.6. RNA Extraction and RT-qPCR

Total RNA was purified from the leaves of the control and treated tomato plants (*n* = 4 per group) five days after spraying using the Spectrum Plant Total RNA kit (Sigma-Aldrich) according to manufacturer’s instructions. cDNA generation and RT-qPCR reactions were performed using the Luna Universal One-Step RT-qPCR kit (NEB, Ipswich, MA, USA) in a Chromo4 operating system (Bio-Rad Laboratories, Hercules, CA, USA). Primer sequences are shown in Appendix A. The correct size of amplified RT-qPCR products was verified by electrophoresis in a 2% agarose gel. RT-qPCR reactions were performed in triplicates. The expression of target genes was normalized to the relevant housekeeping EF1a gene. For each gene, a 2^−ΔCt^ value was measured and used for the calculation of a mean value ±STDEV among all plants per group. Data visualization and statistical analysis were performed using a Student’s *t*-test in the GraphPad Prism software (San Diego, CA, USA).

### 2.7. Bioinformatics and Data Analysis

An analysis of bacterial communities was conducted using Quantitative Insights Into Microbial Ecology 2 (QIIME2) pipeline [26]. Raw reads (.fastq files) were quality trimmed for adapters using the cutadapt plugin [27], joined, and filtered with a minimum quality score of 28. Representative sequences were dereplicated using the *vsearch dereplicate-sequences* tool and clustered into operational taxonomic units (OTUs) at 99% sequence similarity using the open-reference method and the VSEARCH tool [28]. Chimera filtering was performed, and sequences were aligned against the SILVA 132 reference database [29]. Taxonomic classification was performed by filtering out Archaea, chloroplastic, mitochondrial, and unassigned sequences.

Eukaryotes were analysed using mothur according to the provided SOP [30]. Briefly, paired-end reads were joined and trimmed from adaptors and indices. Sequences that could have been generated due to sequencing errors were removed using the *pre.cluster* plugin. Sequences outside the range of 150–450 bp were removed, and the remaining were used for taxonomic assignment into OTUs. The alignment (*align.seqs*) and taxonomic classification (*classify.seqs*) of reads or contigs was performed against the SILVA 132 reference database. A distance matrix was generated to group sequences into OTUs according to their distance value at a cut-off of 0.05. The *remove.lineage* command was also used to filter out Phragmoplastophyta, Archaea, and Bacterial sequences. Finally, chimeric sequences were filtered out using the *chimera.uchime* command.

OTU tables and .biom files were imported in R version 3.6.0 [31] to further process and visualize results. OTU counts and taxonomic assignments were merged to a phyloseq object with the phyloseq R package [32] and analysed using the ampvis2 R package [33]. All plots were visualized by combining functions provided by the ggplot2 R package [34]. All barplots were normalized to 100% as abundance estimations within each sample, so percentages do not reflect the true biomass fraction of each sample.

### 2.8. Quantification of Fungal Load

Microbial DNA isolated from tomato leaves 5 days post-spraying, as described above, was used for the quantification of fungal load by amplifying eukaryotic internal transcribed spacer (ITS) region ITS1 with the ITS primers ITS1F and ITS2. Reactions were performed in triplicate using the Luna Universal qPCR kit (NEB) in a Chromo4 operating system (BioRad). Primer sequences are shown in Appendix A.

## 3. Results

### 3.1. MgO Porous Microparticle Analysis

Table 1 provides an overview of the main properties of the PMP MgO used in this study. Representative images from the TEM analysis of the same PMP MgO, visualizing relevant physical properties of the material, are shown in Figure 1. Commercial MgO NPs have particle sizes in the range of 50–100 nm, while the grain sizes of the MgO PMPs, shown in Figure 1, were in the range of 5–15 nm.

The composition of the powder was 90 wt% MgO, with impurities mostly corresponding to uncalcined MgCO_3_ and Mg(OH)_2_ formed from the reaction with humidity in air (Table 1).

The laser diffraction analysis in Figure 2D showed provided a mean particle size of 5.4 microns, which was orders of magnitude greater than that of commercial NPs (≤100 nm) used in past studies of other research groups. The images obtained by SEM analysis of Figure 2A–C revealed the irregular shape and the composite polycrystalline nature of the particles, as also indicated by the TEM data of Figure 1. The porosity was determined from the BJH analysis, and the surface area was measured using the BET analysis of gas adsorption. The grains, seen in Figure 1, had a mean diameter estimated from radius of gyration of the small angle X-ray scattering curve (SAXS) data of the MgO PMP. The relevant information and analysis will be included in a material-focused manuscript [35].

The XRD analysis shown in Figure 3 is consistent with the composition data of Table 1, showing that the MgO PMPs mainly consisted of MgO and traces of MgCO₃ originating from the mineral sample (Figure 3, black curve).

In sum, the data of Table 1 and Figure 1, Figure 2 and Figure 3 validate the description of the PMP MgO as porous micron-scale particles, composed of nano-grains.

### 3.2. Hydrated MgO PMP Materials

All suspensions used in the current work were 2.3 wt% MgO in water and prepared by mixing deionized water with the MgO PMP in ways to achieve different degrees of hydration of the oxide and according to the descriptions in Section 2.3. For comparison, an additional suspension was prepared with the same MgO powder after its calcination in air at 1100 °C for 2 h. Calcination resulted in a significant loss of its BET specific surface area of 6 m^2^/g, much lower than the 234 m^2^/g of the untreated sample.

The TGA analysis shown in Figure 4 confirmed that the PMP-2 and PMP-3 samples were hydrated substantially, as expected and in agreement with the literature. More specifically, 30–60 min of hydration time for the MgO PMP resulted in >60 wt% Mg(OH)_2_ content, and after approximately 20 h, the content further increased to >70 wt%. In the latter case, hydration temperature did not affect hydration degree, i.e., PMP-2 and PMP-3 were essentially identical in terms of their hydroxide content. If hydration was to be the sole determinant of bioactivity, the PMP-2 and PMP-3 samples would have had the same impact. If the degree of hydration at application was to be a determinant of subsequent bioactivity, the responses of PMP-1 and PMP-3 would have been different.

### 3.3. Tomato Leaf Spraying and Sampling

Leaves from both the water-sprayed control plants and the plants treated with the PMP suspensions were collected from various areas of the plants on the fifth and twelfth days after treatment. The photographs in Figure 5 show that the spray resulted in a non-uniform dispersion of the leaves, with some areas showing white powder accumulation.

This is not uncommon and indicative of a higher dose than the 2 μg/cm^2^ dose used previously in field trials. It is noted that such areas did not show phytotoxicity over the 20 days of the observation period. Moreover, after the experimental period, the plants were transplanted in soil and produced tomatoes. Throughout this time, they remained healthy, not showing any signs of phytotoxicity.

### 3.4. SEM Post-Analysis of Tomato Leaves

Untreated and treated leaves were examined using SEM/EDS in order to assess (a) the homogeneity of active particle deposition, (b) the morphology of the deposition, and (c) the effect of time on the presence of microparticles on the leaf surface (Figure 6). For all PMP-sprayed leaves, EDS analysis verified the presence of magnesium at levels substantially higher than the control leaves, thus confirming the effective application of material on the surface of the leaves (Appendix A).

The SEM images showed that on day 5 after treatment with the PMPs, a relatively uniform coverage of small particles in the range of the initial particles was observed with PMP-1 (PMP1-5d in Figure 6). In contrast, the plants sprayed with PMP-2 and PMP-3 showed more particle agglomerates of various sizes (PMP2-5d and PMP3-5d in Figure 6). In the case of the plants treated with PMP-3, some uncovered areas were also detected.

By day 12, the SEM images showed that the uniform coverage observed on day 5 had almost disappeared from all PMP materials. Some discrete PMP particles or agglomerates were observed along with regions free of particles. The latter suggested that the material was still present but the particles were gradually disappearing from the surface of the leaves.

### 3.5. Assessment of Plant Stress Responses

The visual inspection of the sprayed plants did not show any signs of stress during the post-spraying period (Figure 5), which was consistent with extensive field studies [23] indicating that flash-hydrated MgO PMPs do not induce plant phytotoxicity. To assess the effects of the PMPs on sprayed tomato plants, we tested the induction of the expression of key indicator genes associated with responses to biotic and abiotic stress (e.g., pathogenesis-related protein 2 (PR2), pathogenesis-related protein 2B (PR2B), pathogenesis-related protein 3 (PR3), acidic endochitinase (CHI3), and phenylalanine ammonia-lyase (PAL) genes) [27,28,29,30]. Total RNA and cDNA were prepared from leaves of PMP-water-sprayed and control-water-sprayed plants five days post-treatment. The cDNAs were used in triplicate RT-PCR reactions. As shown in Figure 7A, the PR2 gene was significantly upregulated (~4.3 fold-higher) after spraying with PMP-3 compared to the control group. Similarly, the expression of the PR3 gene was significantly higher in plants treated with PMP-1 and PMP-2 (~3.8 and 3.2 fold-higher, respectively) (Figure 7B). No significant upregulation was observed for the PR2B and CHI3 genes. We also measured the expression levels of phenylalanine ammonia-lyase (PAL), an inducible gene that catalyses the first step in phenylpropanoid biosynthesis [13,19]. PAL expression almost doubled in the leaves of all PMP-sprayed groups, and this increase (~2.4 fold-higher) was statistically significant in the PMP-2-treated plants (Figure 7C). The levels of induction of PR proteins and PAL observed in stressed and infected plants have been documented in other studies to be considerably higher [36], thus indicating that exposure to any of the porous microparticles did not cause major stress in the sprayed plants. However, these levels could function in priming plants to effectively respond to any subsequent insults [1].

### 3.6. Leaf Epiphytic Bacterial Microbiome

To evaluate the leaf epiphytic microbiome community dynamics, upon spraying with magnesium MPs, amplicon metabarcoding analysis was performed using the 16S rRNA and 18S rRNA genes as taxonomic identification markers [37]. The results are shown in Figure 8.

The sequencing of the 16S rRNA gene resulted in 199,984, 195,491, 244,079, and 168,311 raw reads for control dH_2_O, PMP-1, PMP-2, and PMP-3 treatments, respectively. After quality filtering, these numbers were reduced to 157,611, 156,608, 188,216, and 132,647, respectively, ultimately clustering into 54,483 OTUs. Chimeric sequences corresponded to 6.85%, 8.89%, 9.23%, and 4.3% of filtered reads for control dH2O, PMP-1, PMP-2, and PMP-3 treatments, respectively, thus they were removed from downstream analysis. Overall, 30,628 OTUs remained (56.2%) for taxonomic classification. An analysis of the identified filtered OTUs revealed that 835 OTUs were classified as archaea (2 OTUs), chloroplastic (384 OTUs), and mitochondrial (449 OTUs) sequences and were excluded from further analysis. Moreover, 22,938 OTUs could not be classified (unassigned) against SILVA 132 and were also excluded, resulting in 6855 bacterial OTUs to be further evaluated. Of the 6855 OTUs, 754 had a frequency of >1 in total and were used for the analysis. At the order level, Pseudomonadales (86–89%), which consisted of a single Gram (−) *Acinetobacter* sp. genus, dominated the control sterile-water sprayed plants and was represented by 15 dominant OTUs. A far smaller contribution was made by Gram (−) soil bacteria Rhizobiales, as represented by the genus *Allorhizobium* (1–2% abundance); the Caulobacteriales, Gram (−) proteobacteria represented by the genus *Brevundimonas* (1–3%); the Rhodobacterales, coccoid bacteria with nitrate reducing properties represented by the genus *Paracoccus* (1–2%); and Sphingomonadales, Gram (−) chemoheterotrophic aerobic proteobacteria represented by the genus *Sphingomonas*. A total of 265 unique genera were identified in the three sprayed plants, and 10 of them made up 96% of total reads.

In all the PMP-treated plants, a significant decrease in the abundance of the *Acinetobacter* genus occurred (75–0% abundance for PMP-1 and 1–0% for PMP-2 and PMP-3). Gram (+) organisms and cyanobacteria notably increased in abundance. *Sphingomonas* increased 10-fold (5, 8, and 12% for PMP-1, -2, and -3, respectively) from control plants. The Gram (+) extremophile order Deinococcales represented by *Truepera* sp. also increased 10–20 fold (3, 5, and 6%, for PMP-1, -2, and -3, respectively). Gram (+) actinobacteria from the orders Geodermatophiales and Actinomycetales, represented by the genera *Blastococcus* and *Kocuria,* respectively, gained in abundance as well. Cyanobacteria, which were hardly detectable in the control plants, were represented by the genus *Calothrix*, which reached 6% in one plant. In the PMP-1-treated plants, 401 genera were identified, showing that diversity increased as the *Acinetobacter* decreased. The variation of the spread of the orders and genera was not unexpected [38].

In the PMP-2-sprayed plants, *Acinetobacter* was almost fully depleted, with only 25–40 reads per sample present. Several genera filled the void, most of which were also encountered in PMP-1-treated plants. *Sphingomonas* became the most prominent genus (21, 6, and 11% for each replicate), followed by the extremophile Deinococcales *Truepera* (13, 5, and 9% for each replicate). Numerous actinobacteria from the order Geodermatophiales, genera *Blastococcus* (12, 3, and 6% for each replicate) and *Modestobacter* (1, 8, and 1%), were present; an uncultured genus (10, 4, and 11% for each replicate) of the order Rubrobacterales, as well as *Rubrobacter* (5, 1, and 1% for each replicate) and the Actinomycetales of the genus *Kocuria* (1, 9, and 2% for each replicate). Cyanobacteria decreased in abundance compared to PMP-1-treated plants, and 529 genera were identified. In the PMP-3-sprayed plants, *Acinetobacter* was also depleted. The void was covered by the Actinobacteria *Kocuria* (11, 3, and 24% for each replicate) and *Blastococcus* (3, 4, and 2% for each replicate), the Rhodobacterales *Paracoccus* (7, 3, and 10% for each replicate), and the Deinococcales genera *Treupera* (2, 3, and 5% for each replicate) and *Sphingomonas* (3, 7, and 2% for each replicate). Cyanobacteria were represented by the genus *Arthrobacter* (4, 1, and 2% for each replicate). In total, 440 genera were identified.

Principal coordinates analysis (PCoA) was applied to classify and cluster bacterial leaf microbiome samples according to similarities of their identified OTUs for the 12 sprayed plants. Figure 9 shows that the first two principal components (PCo1 and PCo2) accounted for 71.7% of the total genetic variance. Control plants treated with sterile dH_2_O clustered at a great distance from all other treated samples. The PMP-2 and PMP-3 treatments formed two unique subgroups that clustered close to each other, whereas the PMP-1 treatment clustered variably, with two plants falling in between the control and the PMP-2 and PMP-3 subgroups and one plant falling close to the PMP-2 and PMP-3 subgroups.

### 3.7. Leaf Epiphytic Fungal Microbiome

The sequencing of the 18S rRNA gene resulted in 226,240, 206,130, 198,730, and 202,995 raw reads for the control dH_2_O, PMP-1, PMP-2, and PMP-3 treatments, respectively. After trimming, 574,633 reads remained in total, corresponding to 177,420, 129,910, 133,894, and 133,409 reads for the four treatments, respectively, and, after filtering for sequences generated due to sequencing errors, 518,884 sequences remained in total. During chimera filtering, 30,617 sequences were identified as chimeras and were removed, resulting in 488,267 non-chimeric sequences for further evaluation. During taxonomic classification, 450,158 sequences were filtered out, as they were classified as Archaea, Bacteria, or Phragmoplastophyta. Finally, after normalization to the minimum sample size, 1549, 1560, 1563, and 1577 reads remained for the sterile dH_2_O, PMP-1, PMP-2, and PMP-3 treatments, respectively, corresponding to 245 unique fungal OTUs. Of the 245 OTUs, 73 had a frequency of >1 in total. The fungal leaf epiphytome in the control and after the PMP treatment of leaves was dominated by four genera (*Cladosporium* sp., *Dothideomycetes* sp., *Alternaria,* and *Pleosporales*), with two additional genera appearing only in one sample at significant numbers (*Hypocreales* and *Ascomycota*). The treatments did not show significant consistent changes to the fungal profile compared to control samples, in contrast to the bacterial profiles. Even though the fungal genera distribution did not show any major changes between samples, we questioned whether their overall load was affected by treatments. To address this possibility, the ITS region 1 from the leaf surface microbial DNA was amplified by qPCR to obtain a relative quantification of fungal load. The results (Figure 10) indicate that relative to the control leaves, treatment with all PMPs resulted in a significant decrease in fungal load by about 35% for PMP-1, which was increased for PMP-2 (about 50%) and PMP-3 (about 70%), as measured by the relative quantification of the fungal load.

### 3.8. Leaf Epiphytic Bacterial Microbiome in Replicate Experiments in the Fall Season

The amplicon metabarcoding analysis was repeated in the fall season with a new set of *Solanum lycopersicum* L. var. Elpida, commercially grown at that time of the year in Greece. Leaves were collected from six plants; three of them were subsequently sprayed with deionized sterile water as control, and the other three were sprayed with the PMP-2 solution. Nine samples in total, three sprayed with dH_2_O with their corresponding unsprayed counterparts and three sprayed with PMP-2, were used to prepare metagenomics 16S libraries. The sequencing of the 16S rRNA gene resulted in 402,317, 395,896, and 375,315 raw reads for unsprayed, dH_2_O, and PMP-2 conditions, respectively. After all filtering steps, these reads were reduced to 215,871, 221,013, and 206,724, respectively, for the three conditions. In the unsprayed and dH_2_O-sprayed leaves, the dominant genus was *Methylobacterium,* represented by seven distinct OTU, five of them being the species *M. populi* (70, 80, and 53% of unsprayed for each replicate; 72, 82, and 85% of dH_2_O-sprayed for each replicate). Methylobacteria are Gram (−) bacilli of the order Rhizobiales. *Sphingomonas* sp. (two OTUs) ranged from 6 to 15% (7, 6, and 7% for unsprayed for each replicate; 15, 7, and 7% for dH_2_O for each replicate). Another genus belonging to Sphingomonadales, a *Novosphingobium* sp. represented by two OTUs was present in all treatments (6, 2, and 28% unsprayed for each replicate; 4, 3, and 2% dH_2_O; 4, 2, and 3% PMP-2 for each replicate). Treatment with PMP-2 as in the spring experiments caused a drastic shift in the profile of the present bacteria. Methylobacteria decreased dramatically (15, 8, and 9% PMP-2 for each replicate) compared to controlled treatments. *Sphingomonas sp.* was also reduced to 1% abundance. In contrast, two OTUs of DSSD61 belonging to Nitrosomonadaceae, which were largely absent from the control plants, became very abundant (29, 31, and 28% PMP-2 for each replicate). An uncultured cyanobacterium (one OTU) with a low abundance in the control plants was the second most abundant (26, 27, and 24% PMP-2 for each replicate) organism. The extremophile *Acidibacter* sp. (five dominant OTUs) was the third most abundant (12, 17, and 17% PMP-2 for each replicate). *Sediminibacterium* sp. (three dominant OTUs) completed the profile of the PMP-2-treated plants (Figure 10). The PCoA classification and clustering of the microbiome samples from the fall experiment showed that PCoA1 and PCoA2 accounted for 73.8% of total genetic variance. PMP-2-sprayed plants formed a distinct cluster from the control groups (Figure 11).

## 4. Discussion

The objective of this work was to develop an understanding of the bio-activity of flash-hydrated MgO PMPs in agriculture, as seen in field trials as a foliar spray on field tomato crops [23], which have shown efficacy against pests. This work was designed to elucidate the effects of MgO PMPs on the plant and the leaf biome under conditions where the plant had not been impacted by disease. With respect to that impact, the data from the trials were limited, i.e., (a) there was no evidence of phytotoxicity on both short term and long term and (b) there was evidence that beneficial insect populations in the field increased in response to a regular, specified spray program.

Nanoparticles offer the advantage of the effective delivery of agrochemicals due to their large surface area, easy attachment, and fast mass transfer. Metal oxide nanomaterials, such as CuO, ZnO, MgO, and aluminium oxides have been proven in laboratory tests to be effective against foliar and soilborne plant diseases caused by *Botrytis cinerea, Alternaria alternate, Monilinia fructicola, Colletotrichum gloeosporioides, Fusarium solani, Fusarium oxysporum* f.sp. *radicis-lycopersici, Verticillium dahliae, Phytophthora infestans*, and *Ralstonia solanacearum* in many plant species [11,12,13,14].

Commercially purchased MgO NPs were tested by Cai et al. against the Gram-negative, soil-borne *Ralstonia solanacearum*, a pathogen that infects numerous plant species [5]. In liquid suspensions, MgO NPs could completely kill bacteria at a concentration of 250 μg/mL. An examination of the NPs sizes when dispersed in various solutions showed the main group size clustering at 100 nm and a second smaller one at 1 μm. To assess the effects of MgO NPs on the morphology of *R. solanacearum* cells, they examined the microbial cells by SEM. Incubated cells exhibited deep craters on their surface, and many of them were burst. Interestingly, even control bulk MgO had some effects on bacteria, creating small holes on their surface. These visual changes corresponded with increases in the efflux of DNA and proteins into the aqueous solution. A loss of viability was confirmed by flow cytometry. Staining with DCF, an ROS-sensitive dye indicated that MgO NPs contributed to ROS production [5]. An NP suspension in water is expected to convert a portion of them into Mg(OH)_2_, which can directly penetrate cells and cause death. The detailed mode of action of the bioactivity of PMPs is not known. In one possible mode, the high porosity of MgO PMPs bound to the leaf surface allows for the rapid diffusion of bioactive species, such as ROS, through the pores trapped on grain surfaces onto the leaf phyllosphere, such that the overall impact on the plant may be similar to that of the equivalent mass of NPs of such species. In another mode, as the hydration of the PMPs occurs, ROS are generated during hydration from the highly stressed MgO, stabilized on the Mg(OH)_2_ surfaces as MgO_2_ long-lived species, and subsequently released onto the leaf through diffusion. In another possible example case, both PMPs and NPs cause the rupture of cell membranes and can directly release bioactive material such as particle fragment and/or ROS into the cell. The detailed mechanisms for such bioactivity are not documented, and they are likely to be more complex than equivalent NPs [14]. From a chemical perspective, many of the published studies on MgO NPs have not considered the hydration of MgO to Mg(OH)_2_ upon contact with water. For this reason, an aim of this research was to study the impact of hydration of PMPs. The ability of MgO PMPs to create ROS during hydration will be the subject of a upcoming manuscript [35].

In the present study, the starting material was MgO micron-scale porous powder with a high BET surface area 234 m^2^/g, which was equivalent to that of 13 nm dense NPs. Three suspensions were prepared for spraying: PMP-1 was prepared by minimizing hydration before application, while PMP-2 was hydrated over 24 h at ambient temperature, and PMP-3 was hydrated at 90 °C over the same time. PMP-3 was expected to be similar to the flash-hydrated MgO PMP used in the field trials [23]. Tests showed that the degree of hydration was the same for both PMP-2 and PMP-3. The tomato plants were sprayed with these three suspensions, and control plants were sprayed with sterile water. The plants did not show any visible signs of stress or infection throughout our experimentation period of 20 days.

The effects observed on the leaf and its phyllosphere were likely associated with the coating of the PMPs on the plant leaf and the persistence over the term of the experiments. The gradual dissolution of the PMPs was attributed to the absorption of magnesium by the plants. Further work will aim to elucidate the response to the dose rate to monitor and correlate with field trial data.

Previous investigations indicated that MgO NPs induce systemic resistance against the pathogen by activating SA, JA, and ethylene signalling pathways in tomato plants [19]. Aiming to assess the effects of MgO NP application on the tomato plants, we tested the expression of key stress indicator genes. Pathogenesis-related (PR) proteins are mostly small-sized proteins induced by pathogen attack that contribute to resistance to pathogenesis [39,40]. PR1, PR2, and PR5 are induced by SA and have been used as readouts for systemic acquired resistance (SAR) [41]. CHI3 encodes an acidic chitinase whose transcripts are induced in response to insect infestation and priming of plants with a non-pathogenic *Fusarium oxysporum* strain [42,43]. PAL catalyses the first step in the phenylpropanoid pathway [44]. PAL is involved in the biosynthesis of SA, an essential signal involved in plant systemic resistance. PAL gene expression responds to a variety of environmental stresses, including pathogen infection, wounding, nutrient depletion, UV irradiation, and extreme temperatures [45]. The expression analysis of indicator genes in the three PMP-sprayed plants revealed a moderate increase of PR2 and PR3 in specific treatments. In contrast, PAL was induced in all Mg PMP treatments, though at lower levels than those produced naturally by a plant under actual pathogenic attack—an outcome consistent with previous findings with the Mg-NP induction of the SA pathway [19].

The second studied effect was the impact of the PMPs on the leaf bacterial microbiome. The leaf epiphytic and endophytic community is shaped by numerous factors such as host genotype, habitat, growth attributes, climatic factors, soil, and plant symbionts [46]. However, numerous recent studies using next generation sequencing techniques have identified commonalities in the composition of the leaf microbiome. Proteobacteria comprise one such abundant phyla of bacteria present on phyllosphere. In the spring experiments with var. Optima, the most frequent OTUs were assigned to the genera *Acinetobacter, Pseudomonas,* and *Stenotrophomonas*, as well as the *Rhizobia* spp. Minor phyla such as Actinobacteria, Planctomycetes, and Verrucomicrobia are typically also present [47]. A recent examination of the bacterial community of tomato plants (*S. lycopersicum* cultivar “Zhongza 302”) in a greenhouse in China also assigned 97% of phyllosphere reads to the *Acinetobacter* genus [48]. Examinations of the phylloplane from other cultivated plants have also identified striking similarities. In an HTS of microbial community diversity in the soil, grapes, leaves, grape juice, and wine of grapevines from three areas in China, 60–95% of abundance were assigned of *Pseudomonas* and *Acinetobacter* sp. in the leaves [49]. In our experiments, in the water-sprayed control plants, the genus *Acinetobacter* sp. dominated the bacterial community composition (86–89%) in the spring experiments, with minor contributions from *Allorhizobium, Brevundimonas, Paracoccus*, and *Sphingomonas* sp. Treatment with hydrated MgO PMPs caused significant reductions of *Acinetobacter* sp. and the rest of the Gram (−) populations, leaving Gram (+) extremophiles and cyanobacteria that had been previously hardly detectable in the community profile. This change was particularly striking in plants treated with PMP-2 and PMP-3, where five days after treatment, *Acinetobacter* sp. were fully depleted.

In the fall experiments, the most abundant bacterial genera in the phyllosphere of the control plants were *Methylobacterium* sp. They are strictly aerobic, facultative methylotrophic, Gram-negative, rod-shaped bacteria that can grow on one-carbon compounds, as well as on a variety of C2, C3, and C4 substrates [50]. The most abundant OTUs belonged to *M. populi*, a species previously identified in the phyllosphere of poplars. In a similar pattern with the spring experiments, treatment with PMP-2 caused a drastic reduction of the organisms. Extremophiles and cyanobacteria increased in the abundance of identified reads. *Sphingomonadales* present in all samples during both seasons showed a mixed picture, with some OTUs being susceptible and others more tolerant.

Most bacteria in the phyllosphere are harmless, but fresh fruits and vegetables constitute a potential reservoir of opportunistic pathogens [51]. *Acinetobacter* spp. have repeatedly been implicated in nosocomial infections, especially in gravely ill and immunocompromised patients [52]. In a recent study of endophytic bacteria from lettuce and various fruits, the genus *Acinetobacter* was identified in 86% of lettuce and 70% of fruit samples [53]. Further examination at the species level identified a significant percentage of known nosocomial pathogens (*A. baumannii*). The drastic effect that hydrated MgO PMPs have on Gram (−) organisms, which include both human opportunistic bacteria and phytopathogens (*Ralstonia* sp.), therefore has the potential to be a highly beneficial attribute. On the other hand, the elimination of commensals like *Methylobacteria* sp. may be adverse for plant health in the long run.

The third effect is the impact of the PMPs on the leaf fungal microbiome. The fungal population on all tested samples was dominated by three genera, regardless of treatment: *Alternaria* sp., *Cladosporium* sp., and a related uncultured *Dothiodiomycetes*. *Alternaria* is an important plant pathogen believed to cause at least 20% of spoilage losses. This pathogen is known to produce numerous secondary metabolites, many of which are toxic to humans and animals [54]. *Cladosporium* sp. are abundantly found in living and dead plant material. They are encountered as leaf endophytes in plant and tree species [38]. They are highly resistant to adverse conditions. *Cladosporium fulvum* is the causal organism of tomato leaf mould that affects plant foliage [55]. Though the fungal composition was not affected by PMP treatments, the quantification of fungal load on treated leaves showed that PMP treatment coincided with a significant reduction in the density of these fungal pathogens, particularly after treatment with the PMP-2 and PMP-3 hydrated formulations. Our recent observations suggested that the hydrated PMPs provided a fungistatic effect; the suppression of spore germination and mycelial growth rather than a fungicidal effect (unpublished data). Future studies in the field could also address any long-term effects of PMP application on the phyllosphere, seasonality long-term comparisons of plant health and phyllosphere composition, re-colonization of the phyllosphere subsequent to PMP treatments, and changes in the root and leaf endophytic communities and the soil microbiome.

## 5. Conclusions

Previous investigations of MgO and Mg(OH)_2_ nanoparticles had shown them to possess antimicrobial properties. As an economic alternative with the capacity to scale up production, porous micron-size particles (PMPs) were developed and applied to phytoprotection. The grains of these particles are on the 5–15 nm scale, while the particles themselves have mean size of 5.4 microns. The effects of application of differentially hydrated PMPs on the microbial phyllosphere and the leaves of tomato plants were examined. The fully hydrated PMPs caused a dramatic reduction of the dominating Gram (−) bacteria of the phyllosphere, while the overall fungal load was reduced by 50–70%. The plant leaves remained healthy, and an examination of the plant stress response showed only modest increases in PAL gene expression. The results from this study indicate that PMPs produced from hydrated, high surface area MgO powders have great potential for beneficial application in agriculture as a new, non-lethal, and non-toxic active for the suppression of plant pathogens.

## Figures and Tables

**Figure 1 microorganisms-09-01217-f001:**
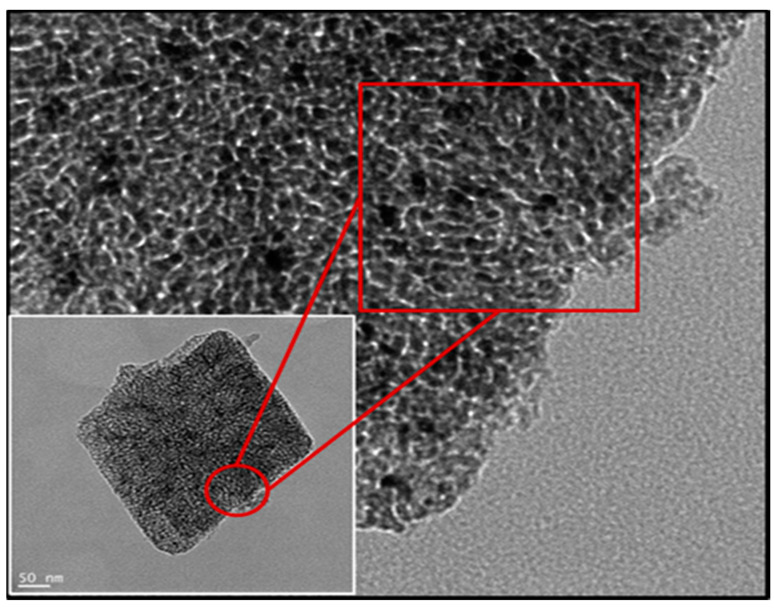
TEM image of an MgO powder particle. In the insert is a photo of a thin particle fragment on the micron scale, and the expanded picture is a higher-resolution image of it that shows the grains on the 5–15 nm scale created by flash-calcination of MgCO_3_.

**Figure 2 microorganisms-09-01217-f002:**
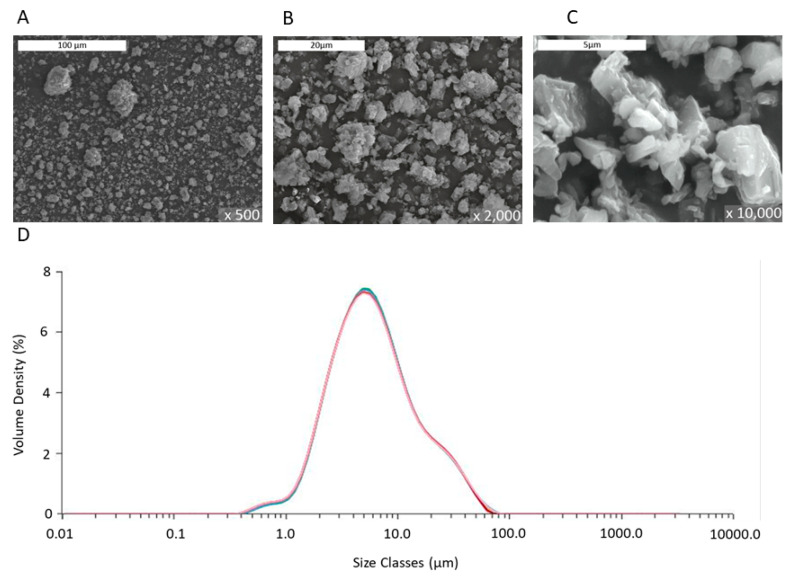
Images of MgO powder studied by SEM at different magnifications: ×500 times (**A**), ×2000 (**B**), and ×10,000 (**C**). (**D**) Particle size distribution of MgO powder by laser diffraction analysis of a dispersion in pure ethanol repeated 10 times (multiple curves shown) to ensure measurement accuracy.

**Figure 3 microorganisms-09-01217-f003:**
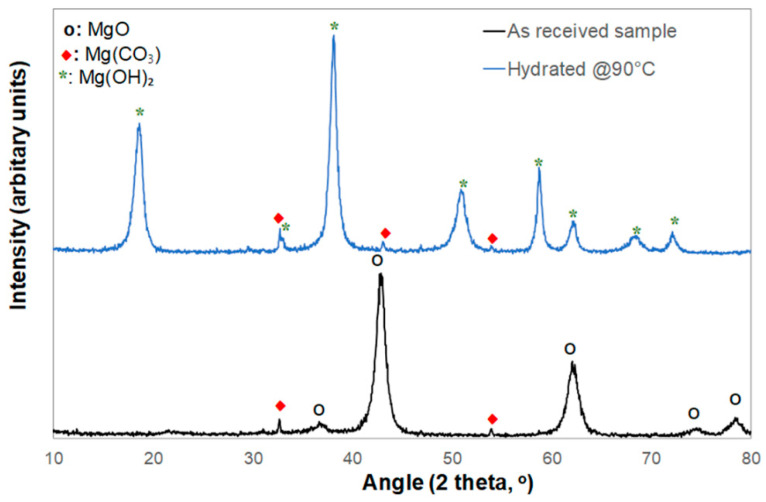
Comparative XRD spectra of the MgO PMP powder as received (black curve) and the Mg(OH)_2_ powder recovered after hydration at 90 °C overnight in deionized water with a typical concentration of 0.04 gr. MgO per mL (blue curve).

**Figure 4 microorganisms-09-01217-f004:**
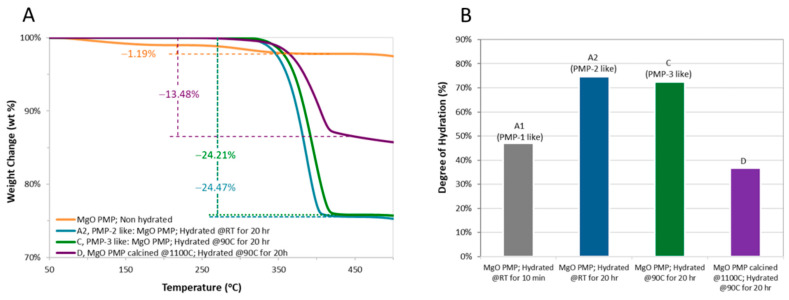
(**A**) Typical weight loss profiles recorded during the TGA of MgO PMP and selected hydrated samples. (**B**) Comparative bar graph, with the Mg(OH)_2_ content estimated from the terminal weight loss recorded in the TGA between 200 and 450 °C. Among the depicted samples, A1 is similar to PMP-1, A2 is similar to PMP-2, and C is similar to PMP-3. The sintered sample (D) hydrated very slowly, reaching a hydroxide content of only about 37% after 20 h. This value was well below the ones of PMP-1 and PMP-3.

**Figure 5 microorganisms-09-01217-f005:**
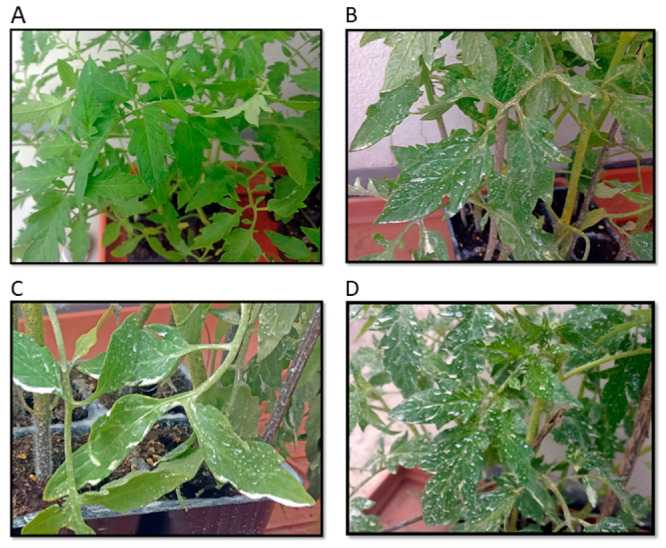
Visual appearance of sprayed plants 5 days post spraying: (**A**) Control, (**B**) PMP-1, (**C**) PMP-2, and (**D**) PMP-3.

**Figure 6 microorganisms-09-01217-f006:**
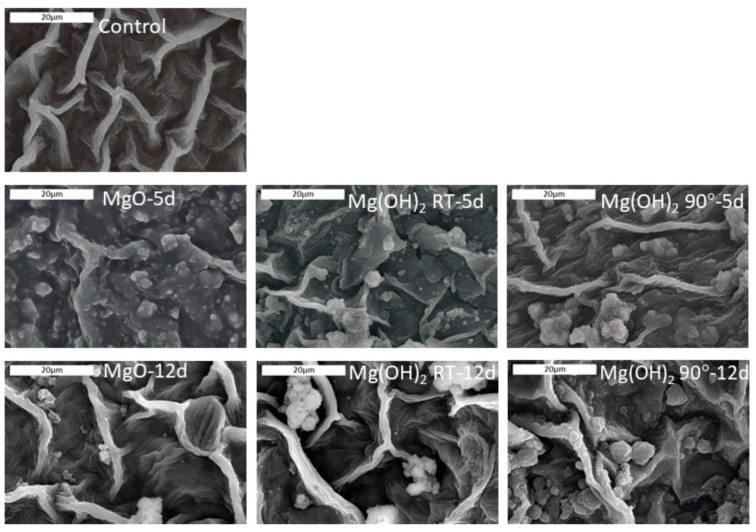
SEM images of the leaves of the control plant (H_2_O) and the plants sampled on day 5 (5 d) and day 12 (12 d) for PMP-1, PMP-2, and PMP-3. The scale bars correspond to 20 μm.

**Figure 7 microorganisms-09-01217-f007:**
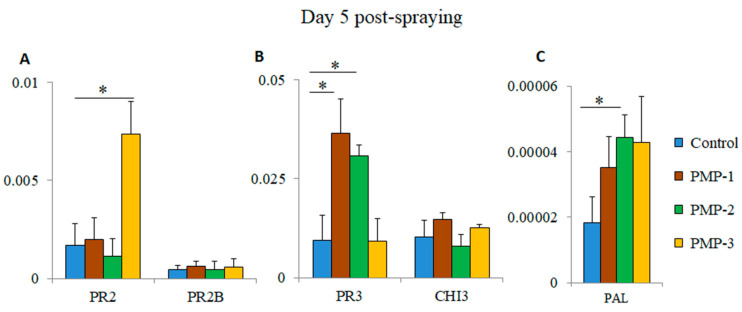
Expression of (**A**) pathogenesis-related protein 2 (PR2) and pathogenesis-related protein 2B (PR2B), (**B**) pathogenesis-related protein 3 (PR3) and acidic endochitinase (CHI3), and (**C**) phenylalanine ammonia-lyase (PAL) genes in the leaves of tomato plants (*n* = 4 per group) 5 days post-spraying with the PMP materials. Error bars denote ± STDEV (* *p*-value < 0.05).

**Figure 8 microorganisms-09-01217-f008:**
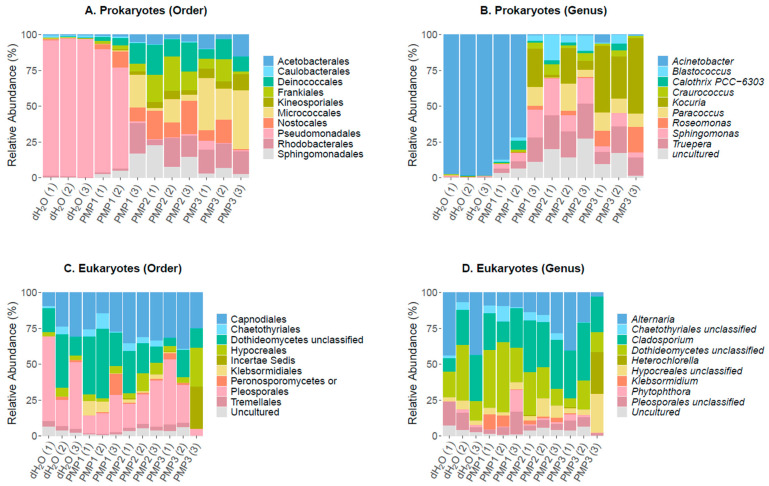
Profile of leaf epiphytic prokaryotic and eukaryotic microbial communities. Distribution of major prokaryotic (**A**) and eukaryotic (**C**) orders and genera (**B**) and (**D**), respectively. The scale in the y axis reflects the normalized relative abundance percentages (%).

**Figure 9 microorganisms-09-01217-f009:**
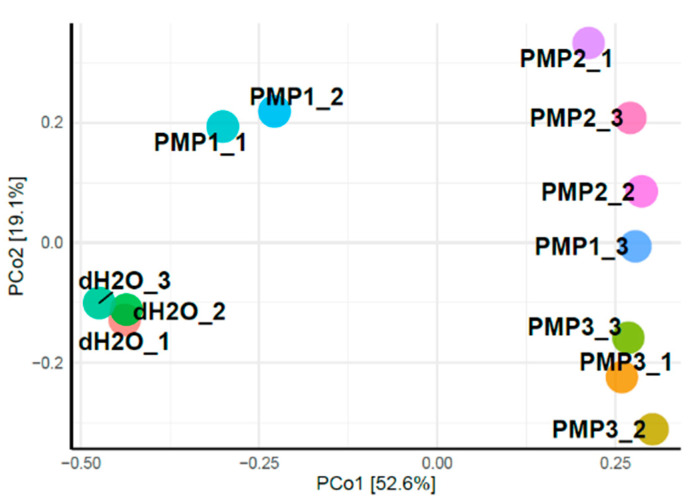
Principal coordinates analysis (PCoA) classified and clustered distinctly bacterial leaf microbiome samples according to similarities of their identified OTUs.

**Figure 10 microorganisms-09-01217-f010:**
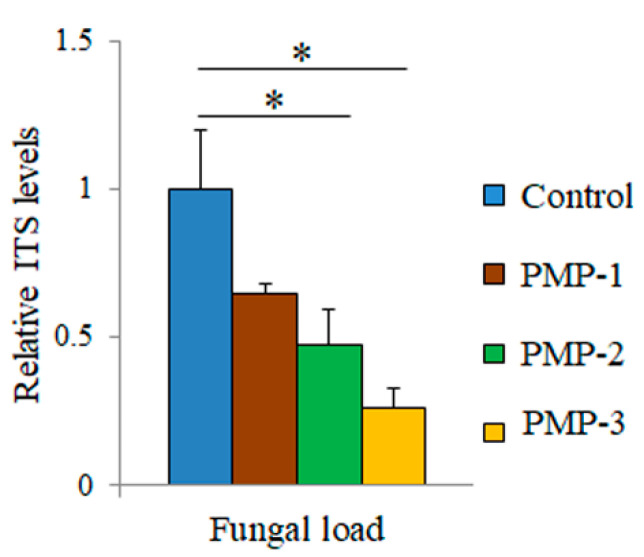
Quantification of fungal load in the leaves of tomato plants 5 days post-spraying with microparticles prepared from porous microparticle MgO. ITS levels in control samples were arbitrarily set to 1. Error bars denote ± STDEV (* *p*-value < 0.05).

**Figure 11 microorganisms-09-01217-f011:**
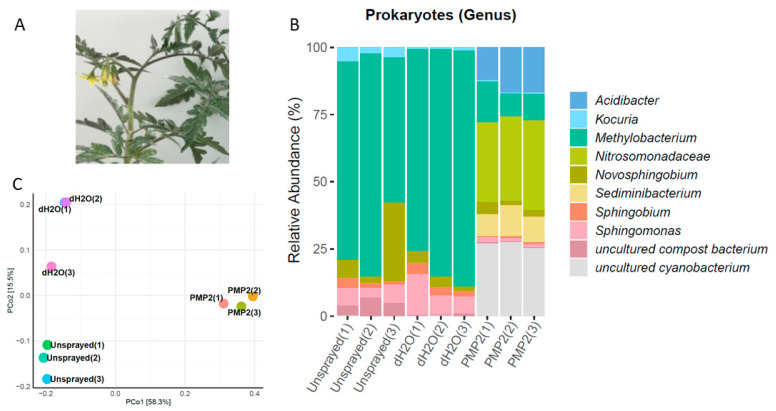
(**A**) Visual appearance of PMP-2-sprayed plants 5 days post spraying in the fall experiment. (**B**) Profile of leaf epiphytic prokaryotic genera microbial communities. The scale in the y-axis reflects the normalized relative abundance percentages. (**C**) PCoA classified and clustered bacterial leaf microbiome samples according to similarities of their identified OTUs.

**Table 1 microorganisms-09-01217-t001:** Physical properties of MgO porous micron-sized particles.

Chemical Composition wt%		Physical Properties	
MgO	90.03	Particle Size	d_10_ 1.96 μm
MgCO_3_	6.53		d_50_ 5.37 μm
CaMg(CO_3_)_2_	2.55		d_90_ 16.80 μm
CaCO_3_	0.24	Porosity	0.47
CaAl_2_Si_2_O_8_	0.47	Surface Area	234 $\;m^{2}/g$
Other (traces of Fe_2_O_3_, SiO_2_ etc.)	0.09	Grain Size	5–15 nm
	100%		

## Data Availability

Sequencing data have been deposited in NCBI at BioProject ID PRJNA638533.

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
