# Peer review of "Effects of Magnesium Oxide and Magnesium Hydroxide Microparticle Foliar Treatment on Tomato PR Gene Expression and Leaf Microbiome"

_microorganisms, 2021, doi:10.3390/microorganisms9061217_

Round 1

Reviewer 1 Report

The comments refer to the work microorganisms-1208996 original and the line indications refer to it.

L31-35: this part does not fit in at all with the theme of the work, the introduction could easily begin with L35 and a description of the crop losses.

L48-51: references?

L56: unnecessary insertion (reviewed by Shang et al. [13]).

L56-57: „differing models of the mode of action of NPs” – it is worth mentioning some.

L62: references?

L63-64: Why do you only refer to wheat that is not work-related?

L77: space 250 μg/ml

L98: correct unit record 20 microgram/cm2 – 20 μg/cm2

L104-107: a lot of methodological details, they don't have to be here at work

L112-113: maybe not a proposition but a hypothesis? A research hypothesis is worthwhile.

L144: space 100 ml/min,

L129-146: references? Or author's methods? Analyses done in replicates?

L153: correct unit record 2.2 microgram/cm2 – 2.2 μg/cm2

L164: space < 3 min

L208-216: it would be clearer to present sequences, references and primer names in a table

Bioinformatics and data anlaysis subsection (2.6) should be after RNA extraction and RT-qPCR (2.7), because some analyses were used here too.

L305: space 100 nm

Table 1: not formatted according to, not clear. What does other mean?

L279: space 90 wt%

Figure 2D: totally unreadable

Figure 3: poor quality

L323, 324: space > 60 wt% and > 70 wt%.

Figure 4: totally illegible, instead of "top"/"bottom" should be A and B.

L345-346: these abbreviations should be excluded under figure

Figure 7: poor quality, no explanation of abbreviations, what test used to determine differences and what was n?

L403-405: percentages should have dots between digits, not commas

Figure 8: totally illegible

Figure 11: illegible, instead of "left"/"centre"/"right" should be A and B. Blurred fonts

L534: unnecessary insertion (reviewed by Shang et al. [13]).

L538-540: space 250 μg/ml; 100 nm, 1 μm

L547-573: too few references to other studies

Author Response

Response in the attached file

Reviewer 2 Report

I have no further comments. Thanks to the authors for corrections and improvements to the article.

Author Response

We would like to thank Reviewer 2

Round 2

Reviewer 1 Report

Thank you for your corrections.

I accept the paper.

This manuscript is a resubmission of an earlier submission. The following is a list of the peer review reports and author responses from that submission.

Round 1

Reviewer 1 Report

This manuscript describes the results of a study into the impacts that foliar application of magnesium microparticles has on leaf gene expression and surface-associated microbiota of tomato plants.

One of the main conclusions of the paper is that such an application causes a shift in the leaf surface microbiota ‘from near 100% dominance by Gram (-) bacteria, leaving Gram (+) extremophiles and cyanobacteria to cover the void’ (lines 20-21). This finding is presented as one of the main impacts of Mg microparticle application. Other impacts are more ‘modest’ (gene expression) or absent altogether (fungal microbiota).

Closer inspection of the bacterial data (Figure 8) reveals that all (three) samples of the untreated control consist almost entirely of Acinetobacter species, as do most (two out of three) samples of one of the treatments. This type of dominance by a single OTU at the genus level is atypical of phyllosphere microbiota. An alternative explanation is that this finding represents a contamination with Acinetobacter, which would skew the results. To rule out this possibility, the experiment needs to be repeated at least two more times independently (the data in the manuscript are from one single experiment only). Without such repetition, the conclusion as presented in the abstract and paper is not justified.

The manuscript lacks details on some of the experimental procedures. It says that there are three plants per treatment, but how many leaves were collected and pooled (or not) from each one of those plants? For gene expression analysis, was the RNA collected from different leaves? Also, from how many leaves from each one of the plants was RNA collected?

Other comments

Line 20: ‘phylloplane’ is used incorrectly here: replace with ‘phylloplane microbiota’.

Line 21: ‘cover the void’: this is conjecture: there is no evidence shown that supports the existence of a void.

Line 34-36: Any such data available for tomato, the plant under study here?

Line 50: ‘plant and soilborne diseases’: please clarify whether this means ‘plant diseases that are soilborne’ only.

Lines 51 and 435: check proper way to write ‘fsp’. Also, ‘Radicis’ is not capitalized.

Lines 55-70: it is unclear what the connection is of this paragraph with all else in the manuscript. I suggest to remove.

Line 99: remove ‘at’.

Line 104: ‘spaying’.

Line 179: Unclear what this means, please rephrase: ‘On the fifth and the day of spraying’.

Line 182: The 10 min and 2,683xg settings do not sound like they are sufficiently long or high, respectively, to collect all microorganisms from the 40 ml leaf wash. Can the authors comment on this? Also, does the powder on the treated plants contribute to greater retrieval of microorganisms using this method of centrifugation. Did the authors test for this possibility?

Line 185: provide data or reference for this statement.

Figure 1: Explain what is shown here: is that one powder particle in the left bottom corner? Confusing, because the legend talks of ‘particles’ (plural).

Line 307: Specify how much higher.

Line 308: Mention whether there was any phytotoxicity beyond 20 days.

Figure 6: It is difficult to make judgment from these pictures, each one of which is the sole representative of a single treatment. Is there a way to complement with a more quantitative assessment of these data?

Figure 7: It is not possible to interpret this figure without a label on the y-axis.

Lines 429-436: This section feels repetitive (lot of overlap with Introduction). Same to some degree for lines 437-461.

Reviewer 2 Report

In principle, the article is not bad, it is written clearly, although some passages are repeated (see below, I note this), the pictures are also understandable and well presented. Nanoparticles are a current topic and it is good that the authors focus not only on the effect on plants but mainly on the microbiome. However, I would expect a much broader discussion here. Instead of recurring passages, authors can use this space to broaden the discussion of NGS data

  • Line 104 - spaying aqueous > spraying aqueous
  • Line 174 - plants (Solanum lycopersicum) > (Solanum lycopersicum L.) please specify species varietas
  • How the authors determined the amount of sprayed nanoparticles.
  • It is a generally accepted truth that PAL genes may indicate the stress conditions of a given organism. pal is considered a divide between primary and secondary metabolism. Why was the PAL 4 gene selected from all PAL genes
  • There is a bit of a confusing statement in the introduction where it is written first about reducing insect populations and then in the discussion it says that the insect population has grown. Please explain these sentences. ("he foliar application of PMP suspensions resulted in at about 20 microgram / cm2 leaf coverage to elicit either a reduction in damaged fruit, associated with reduced insect pest pressure or, a reduction in the frequency at which pest pressure reached economically damaging levels. Such PMP treatment required substantially less insecticide and fungicide compared to control plots, which were farmed using conventional crop protection actives alone ") and expression in a discussion (" With respect to that impact, the data from the trials was limited, namely (a) there was no evidence of phytotoxicity on both short term and long term, and ( b) there was evidence that beneficial insect populations in the field increased in response to a regular specified spray program ").
  • LINE 46 and 47 introduction and line 429 and 430 discussions are same please remove it or rewrite. “46 Nanoparticles (NPs) are organic, inorganic or hybrid particles with at least one of their dimensions ranging 47 from 1 to 100 nm (at the nanoscale).” “429 Nanoparticles (NPs) are organic, inorganic or hybrid materials with at least one of their dimensions ranging 430 from 1 to 100 nm (at the nanoscale)”
  • The whole paragraph is needed to rewrite. Please do not use the same text in discussion and introduction “Nanoparticles (NPs) are organic, inorganic or hybrid materials with at least one of their 430 dimensions ranging from 1 to 100 nm (at the nanoscale). Nanoparticles offer the advantage of  431 effective delivery of agrochemicals due to their large surface area, easy attachment and fast mass  432 transfer. Metal oxide nanomaterials, such as CuO, ZnO, MgO and aluminum oxides have been  433 proven in laboratory tests to be effective against many plant and soil borne diseases caused by Botrytis  434 cinerea, Alternaria alternate, Monilinia fructicola, Colletotrichum gloeosporioides, Fusarium solani, Fusarium 435 oxysporum fsp Radicis lycopersici, Verticillium dahliae, Phytophthora infestans and Ralstonia solanacearum  436 in many plant species [9-11] (reviewed by Shang et al. [12]).”
  • The authors obtained a lot of beautiful data, but would like a more detailed description, especially from NGS.
  • I would like to mention that these nanoparticles have a negative impact on bacteria and fungi in this case dangerous to the plant. However, it would be good to mention or show how PMP affects symbiotic fungi and rhizobacteria, which are an integral and necessary part of plants as well as soil microflora. I understand that nanotechnology or nanoparticles can be a new generation of pesticides, but it is important to know their effect not only on plants but also on the soil microflora.
  • Could be good to mention in the article microbiomes of soil or at least CO2 patterns of soil treated with individual PMPs.
  • I miss the conclusion in the article.